# Microorganism Diversity Found in *Blatta orientalis* L. (Blattodea: Blattidae) Cuticle and Gut Collected in Urban Environments

**DOI:** 10.3390/insects15110903

**Published:** 2024-11-19

**Authors:** Constanza Schapheer, Luciano Matías González, Cristian Villagra

**Affiliations:** 1Departamento de Ingeniería y Suelos, Facultad de Ciencias Agronómicas, Universidad de Chile, Santiago 8820808, Chile; 2Instituto de Entomología, Facultad de Ciencias Básicas, Universidad Metropolitana de Ciencias de la Educación, Santiago 7760197, Chile; lgonzalezv3@correo.uss.cl

**Keywords:** pathogen, biotic homogenization, ectomosphere, holobiont, nosocomial diseases, public health

## Abstract

Urban cockroaches raise health concerns due to their association with pathogens. However, their microbial associates can enhance their physiological and reproductive functions. This article details the bacterial community associated with the oriental cockroach *Blatta orientalis* Linnaeus, 1758 (Blattodea: Blattidae) using metabarcoding for the first time. We analyze bacterial communities on the exoskeleton and within the gut of this cosmopolitan pest. Specimens were collected from the urban area of Santiago, Chile. DNA extraction and metabarcoding were performed for analysis. Our findings reveal a variety of bacterial lineages, including mutualistic symbionts and pathogenic strains. We examined the metabolic functions of these bacterial communities and their implications for *B. orientalis* as a pathogen reservoir and vector of zoonosis. Lastly, we consider how the microbial diversity in cockroaches might facilitate their adaptation to human-altered environments.

## 1. Introduction

Insects inhabiting human-modified habitats, such as cockroaches and house flies [1,2,3], can become vectors of several diseases as their lifestyle puts them in direct contact with humans, their resources, and derived wastes [4,5]. Through this proximity, microorganisms can colonize the body of animal vectors [6]. Once microorganisms colonize the body of these vectors, they find a “mobile refuge” that often protects them from chemical and pharmacologic control exposure until they can reach their human hosts [7,8]. In cockroaches (Blattodea), it has been demonstrated that microorganism communities (pathogenic and non-pathogenic) can be hosted inside internal organs, e.g., the digestive system, as well as over the surface of the exoskeleton cuticle, also known as “ectomosphere”. These different body areas can be considered fine environmental grain, providing proper niches for different pathogenic microorganisms [6]. Thus, their study may allow us to disentangle their potential contribution to microorganism diversity vectored by urban pest cockroaches [9,10,11]. 

Bacteria play fundamental biological functions in most insect lineages, participating in the co-construction of their trophic and reproductive characteristics [12]. In synanthropic insects, mutualistic symbiotic bacterial strains may contribute in many ways to their survival and success in human-modified environments [13]. In pest cockroaches, with the exception of a few well-studied pathogenic bacteria, gut microbiomes have revealed many lesser-studied bacteria lineages related to physiological and immunological features [14]. Therefore, it would be relevant to consider the diversity of bacterial associates as a whole, not only those causing diseases in humans, when addressing the roles of these partners in pest insect biology [15].

For cockroaches, the interaction with bacteria has been proposed to evolve in several instances of transition from parasitism to symbiosis [16]. The associations of Blattodea with bacteria can reach intricate dynamics at a cellular level, such as the case of intracellular bacteria symbionts (endosymbionts) hosted on insect’s specialized cells, as it has been described for the oriental cockroach, *Blatta orientalis* L. (Blattodea: Blattidae) [17]. This is the case of *Blattabacterium cuenoti,* a primal endosymbiont of this cosmopolitan pest, inhabiting specialized abdominal fat body cells [18]. Moreover, besides being hosted inside the insect organs, bacterial communities can also be in the outer layer of the cuticle or ectomosphere of this cockroach. Regarding research on bacteria associated with *B. orientalis*, a vast majority focuses on the search for strains pathogenic to humans, leaving aside other possible insect-microorganism associations such as mutualistic symbiotic bacteria lineages. For example, it has been described several pathogenic bacteria lineages, such as *Enterobacter* and *Salmonella* [19,20,21] (Table 1; [19,22,23,24]).

In this work, we collected *B. orientalis* in Santiago, the capital city of Chile. We extracted microorganism communities from this insect’s gut and exoskeleton ectomosphere. We conducted molecular extraction and sequencing to explore the microorganism’s diversity composition and the preponderance of non-pathogenic (e.g., mutualistic symbionts) and strains pathogenic to humans found in this synanthropic insect. We found a considerable fraction of non-pathogenic microorganisms in these pest insects, while other strains that may function as medically concerning microorganisms corresponded mainly to *Erysipelatoclostridium, Enterococcus,* and *Sphingomonas*. We discuss the diversity of the bacteria and their association with these pest insects, their relevance in this synanthropic cockroach life cycle, and the potential hazards implied in the presence of the few pathogenic lineages detected. 

## 2. Materials and Methods

### 2.1. Study Sites and Species Determination

In the summer of 2020 in the southern hemisphere (January–February), we collected *B. orientalis* specimens (n = 6) in Santiago, Chile. This is the most prominent urban nucleus in this country, corresponding to the seventh largest city in the Americas, with over 7,112,808 inhabitants counted in the last census of 2017 [25]. We captured cockroaches in urban sites within Santiago de Chile (Figure 1): Site 1 (San Miguel; 33°30′23.4″ S 70°38′52.5″ W), three adult males and one adult female; and Site 2 (Vitacura; 33°21′57.2″ S 70°33′13.3″ W), one adult male and one adult female. In both cases, the cockroaches were collected in the backyards of houses with owners’ permission through the active catching of adult specimens using sterile globes and sterile centrifuge tubes (25 mL volume) as containers. We used a general key for cosmopolitan cockroaches and pests present in Chile (extracted and modified from [26,27]) for taxonomic determination. 

### 2.2. DNA Extractions and Amplification

Cockroach guts: For extracting the microorganisms hosted inside the guts of the insects, first we washed each cockroach three times for two minutes with a 0.5% NaClO solution, then rinsed with 70% ethanol, and finally three times for three minutes with sterilized water to remove the microbiota from the cockroach’s cuticle [28]. We then dissected them using sterile forceps, removing the entire intestine, from which we extracted the DNA. Thus, samples used for these extractions cannot be also utilized for extracting the microbiome associated with insect ectomosphere. In this case, we used four specimens for gut analysis: two females and two males, one from each site for both sexes. 

Cockroach cuticle: We used two male specimens from Site 1 for this analysis. We placed each cockroach in a 50 mL tube with a wash solution of 0.9% NaCl, 0.02% polysorbate 20, and distilled water. The tubes were left for 30 min on a Mx-Rd-Pro disk rotator at 70 rpm DLAB^®^ (Beijing, China). Then we extracted the insect, taking care not to break it, and centrifuged the solution at 15,000 rpm for 15 min (D2012 DLAB^®^). DNA was extracted from the resulting pellet. In both cases, the DNeasy PowerSoil Pro Kit (Qiagen^®^, Hilden, Germany) extraction kit was used for DNA extraction, following the manufacturer’s specifications. The quantity and quality of the extracted DNA were quantified by fluorimetry using a Qubit 4.0^®^ DNA quantifier (Thermo Fisher Scientific, Waltham, MA, USA). 

We amplified the V3-V4 variable region of the 16S rRNA (464 bp) gene using 341F (CCTACGGGNGGCWGCAG) and 805R (GACTACHVGGGTATCTAATCC) primers [29]. Illumina sequencing was performed at Genoma Mayor, Universidad Mayor, Chile, following the protocol based on “Illumina 16S Metagenomic Sequencing Library Preparation”.

### 2.3. Bioinformatics and ASV Inference

Trimming and filtering were performed on the data obtained from Illumina sequencing, based on inspection of the quality graphs, with a cut-off criterion of a maximum of two errors per paired-end reads. We used the DADA2 [30] bioinformatics protocol on the R platform for sample analysis. The steps were as follows: (1) concatenation of paired reads, (2) chimeric sequences removal, (3) estimation of the sequencing error per base, (4) estimation of the replication error (i.e., elimination of redundant sequences), and (5) inference of ASVs. We completed taxonomic assignment via a native implementation of the Naives RDP Bayesian classifier [31], and 16S rDNA gene sequence assignment by exact matching with the SILVA v132 database [32]; we removed sequences corresponding to non-bacterial lineages. We characterized the core microbiome at the genus level, considering a minimum detection threshold of 2% and a minimum prevalence of 60%. We made a heatmap to illustrate the abundance of the main genera of bacteria and how they are grouped according to sample type. To determine the functionality of the microbial communities, we predicted functional pathways using the KEGG ortholog (KO) database [33]. For data visualization, we used MicrobiomeAnalyst 2.0 (https://www.microbiomeanalyst.ca/; accessed on 12 September 2024) (https://www.microbiomeanalyst.ca/, accessed on 12 September 2024) [34,35,36].

## 3. Results

### 3.1. Diversity and Abundance of Bacterial Communities

The bacterial community of *B. orientalis*’ gut and cuticle were determined, and 4671 ASVs were assigned (Appendix A). From described bacteria from previous works, most lineages were collected from inside the cockroach gut, while a reduced number of reported strains corresponded to its ectomosphere (e.g., cuticle) (Table 1). Furthermore, most previously reported bacteria were pathogenic to humans. In this work, we explored external and internal insect areas as fine-grain bacteria environments, considering both pathogenic and non-pathogenic groups (e.g., mutualistic symbionts) that may contribute to *B. orientalis* lifestyle success as an urban pest. 

The most abundant phyla for both gut and cuticle corresponded to Firmicutes, Bacteroidetes, and Proteobacteria; at the genus level, *Lactobacillus* and *Bacteroides* were among the most abundant (Figure 2).

When the intestines and cuticle were examined separately, we observed that the most abundant genera, *Lactobacillus* (33%), *Bacteroides*, (22%), and *Blattabacterium* (13%), are also those that conform the averaged core microbiome in the case of the gut (n = 4). While, in the case of the cuticle, based on the sample corresponding to two males of *B. orientalis*, there is ample variation in the genera that make up the core microbiome (Figure 3). In one male, the most abundant corresponded to *Enterococcus* (31%), *Acinetobacter* (12%), and *Lactobacillus* (12%), while for the second male cuticle sample, this corresponded to *Acetobacter* (66%), *Sphingomonas* (21%) and *Bacteroides* (2%) (Figure 3). This contrast between samples can also be observed in Figure 1. 

The heat map developed shows a separation in two distinctive groups of bacteria found in *B. orientalis* intestines and ectomosphere (Figure 4). Sample type (gut or cuticle) helps to discriminate the bacterial diversity from these pest insects.

### 3.2. Kyoto Encyclopedia of Genes and Genomes (KEGG) Analysis

After the KEGG analysis, we found 1945 KO (KEGG Orthology) associated with a metabolic function (Appendix A) in the total of the samples. The representative abundance of metabolic functions grouped based on sample type according to KEGG prediction is shown in Figure 4. Regarding the inference of the functions of bacterial communities, those that are most expressed in gut(G) and cuticle(C) are amino acid (18.2%G and 18.8%C), carbohydrate (20.1%G and 18.6%C), and energy nucleotide metabolism (17.8%G and 18.7%C) (Appendix A; Figure 5).

## 4. Discussion

In this research, we present the first detailed molecular study that comprehensively characterizes the bacterial community of *B. orientalis*. We considered both ectomosphere and gut body areas that are distinct microhabitats for pathogenic and non-pathogenic bacterial communities associated with the oriental cockroach. Traditionally, the bacteria associated with cockroaches were studied with culture techniques where mainly pathogenic bacteria were isolated; however, with molecular techniques, it is possible to have a more global idea of the entire community accompanying the insect [14]. Previous works reported the genera of human pathogenic bacteria: *Escherichia*, *Proteus*, *Micrococcus*, *Aerococcus*, *Hafnia*, and *Morganella* (Table 1). However, they did not appear in our results (Appendix A). 

Regarding lineages pathogenic to humans, we found in *B. orientalis* core microbiome, and we detected *Erysipelatoclostridium* in cuticles and digestive tracts from cockroaches collected in Santiago (Figure 1 and Figure 2; Table 2; [23,37,38,39,40,41,42,43,44,45,46,47,48,49,50,51,52,53,54,55,56,57,58,59,60,61,62,63,64,65,66,67]). 

Within this genus, bacteria such as *Erysipelatoclostridium ramosum* are regarded as the source of several human infectious conditions in immunocompromised patients [51]. This genus is reported among pathogens found in pest cockroaches; however, vectoring dynamics have been barely been explored for this particular group [68]. We also found species from the enteric bacteria genus *Enterococcus* in the cuticles and guts of *B. orientalis* from Santiago, Chile. In previous work, this group was reported for the German cockroach (*Blattella germanica* L.) guts, where these strains do not trigger disease responses to the host insect [69]. Furthermore, we found several *Sphingomonas* only in *B. orientalis* cuticles (Figure 2 and Figure 3, Table 2). Within this group are several potential virulent bacterial pathogens for humans [70]; nonetheless, to our knowledge, this is the first time this genus has been found in cockroaches.

Besides these core bacterium lineages, there are potentially pathogenic groups that are less represented, such as *Staphylococcus* (Appendix A). In previous work with *B. germanica,* both in their ectomosphere and inside their guts, it was possible to find *Staphylococcus aureus* on their body surfaces and within their digestive tracts. This strain is a normal component of the human microbiome but can also trigger virulent infections [71]. Cockroaches can be both reservoirs (harbor) and vectors (spread) of various foodborne pathogenic bacteria, such as *S. aureus*, capable of producing toxins triggering food intoxication [71]. This process may require just a small amount of pathogenic bacterium and its inoculation on food sources with the help of their insect partners [72]. Thus, this insect bacteria association poses a public health risk, especially in hospitals where they can function as vectors for nosocomial pathogens [73]. The detection of pathogenic lineages must be completed by integrating specialized techniques, such as selective cultures and molecular analysis, thus allowing for a precise determination applicable to different contexts (Table 1; [74]). A complete list of potentially pathogenic ASVs extracted from *B. orientalis* cuticle surface and guts can be found in Appendix A.

Concerning microorganisms non-pathogenic to humans found in this study, we discovered that bacteria constitute the majority of the community and core microbiome of the gut, highlighting the phyla Firmicutes (e.g., *Lactobacillus*) and Bacteroidetes (e.g., *Bacteroides and Blattabacterium*) (Table 2, Figure 2 and Figure 3). Several of these ASVs were not reported in previous works on other pest cockroach species (Tables Table 1, Appendix A), but are regarded as part of different organisms’ microbiomes with diverse functions, not only pathogens (Table 2). Within Bacteroides for instance, several strains are described as part of animal microbiota, participating in gut health and nutrient assimilation [42]. *Bacteroides* symbionts have been found as relevant players for the adaptation of insects to the use of different dietary resources, including polysaccharides, where, thanks to *Bacteroides* polysaccharide utilization loci (PULs) and carbohydrate-active enzyme (CAZyme) coding genes, their insect biont counterparts are capable of dealing with complex polysaccharides (e.g., starch, pectin, and cellulose), boosting nutrient availability for this holobiont partnership. This has been reported for another Blattidae pest cockroach: *Periplaneta americana* L. [75]. This is consistent with our functional inference analysis of the bacterial communities and the fact that the most highly expressed function was carbohydrate metabolism (Appendix A; Figure 5).

In the diverse gram-positive genus *Lactobacillus* (Lactobacillaceae), there are several examples of their preponderance as endosymbionts in different insect lineages, such as Hymenoptera, Diptera and Blattodea, where it is suggested as key in their trophic diversification [76,77,78]. In bees (Hymenoptera: Apidae), these lactic acid bacteria strains are hosted in the digestive tract, being part of the core gut microbiomes of different species [77,79]. These bacteria present adaptations that facilitate them to prosper in the nutrient-rich *milieus* of the bee digestive tract and in the plant’s floral structures rich in nectar and pollen resources [77,80]. Regarding symbiosis with cockroaches, *Lactobacillus* is present in the digestive microbiome of the brown-banded cockroach, *Supella longipalpa* (Fabricius, 1799) (Blattodea: Ectobiidae), where it contributes to its biont nutrition and development [78]. 

In the case of cuticles, the most abundant phyla were Firmicutes and Proteobacteria, which are also shared with the digestive tract community (Figure 2 and Figure 3). Moreover, in less abundance, but only in *B. orientalis*’ ectomosphere, we found Enterobacteriaceae (e.g., *Shimwellia*), Moraxellaceae (e.g., *Acinetobacter*), and Ruminococcaceae (e.g., *Novosphingobium*), among others. Inside the digestive tract, core bacteria were dominated by gram-negative anaerobic Bacteriodetes and Firmicutes, with a predominance of families such as Lactobacillaceae (e.g., *Lactobacillus*) and Bacteroidaceae (e.g., *Bacteroides*) (Table 2, Figure 2 and Figure 3). In this insect-borne microorganism habitat, lineages like Desulfobacteraceae (e.g., *Desulfobotulus* and *Desulfovibrio*) and Oscillospiraceae (e.g., *Anaerofilum*) were exclusive for gut samples. This entire network of interactions could partly explain the way of life of cockroaches. From the perspective of the ecosystemic holobiont proposed by Schapheer et al., 2021 [12], it is of utmost relevance to understand the emergent properties that underlie the attributes of the insect both in a pest management context and in a wild habitat.

This study did not find *Wolbachia* (Appendix A), a widespread endosymbiotic bacterium in insects. This lineage has been reported as relatively uncommon, both in pest and native cockroach species [81,82]. In ongoing research, we obtained this group on native lineage *Moluchia* Rehn, 1933 (Blattodea: Ectobiidae) using the same protocol (C. Schapheer, personal communication). Other authors have also found *Wolbachia* in reduced percentages in the Ectobiidae, Blattellidae, and Blaberidae families [81,82]. Moreover, to our knowledge, surveys on Blattidae species, such as *P. americana*, have not found individuals bearing *Wolbachia* strains [81]. Thus, this evidence is coherent bacteria from this group was not found for *B. orientalis* in our study. It is possible that this group is absent in this synanthropic pest. However, further and extended sampling is required to confirm. 

Like most insects, cockroaches depend strongly on their microorganism associates; these partners influence their lifestyle and phenotypic features [12,83]. Moreover, a portion of their microorganism’s load may correspond to pathogens capable of inflicting several diseases when transferred to suitable hosts like humans. Cockroaches in cities may be capable of mobilizing pathogenic strains from contaminated sources to novel human hosts [20,84]. In addition, they can actively escape insecticide and hygienic and pharmacologic control measures with their microorganism cargo [85], keeping these strains exposed to low doses of these chemical formulations that may be capable of triggering hormetic responses and resistance [86,87]. Considering all of the above, we must mention that the results obtained in this research should be taken cautiously, considering the limitations of the sampling and the small sample size. We hope that future research will delve deeper into the community of bacteria and other microorganisms associated with *B. orientalis.*

## 5. Conclusions

In this work, we found that only a low portion of the bacteria associated with urban *B. orientalis* collected in Santiago corresponds to human pathogens. The rest of the associated microorganisms would be part of the normal microbiota of the insect. However, the effect of *B. orientalis* as a reservoir and vector of microorganisms of medical interest should not be underestimated since, in the case of highly pathogenic bacteria, a small amount is needed to cause disease.

Among urban cockroaches, *B. orientalis* has been highlighted by its elevated vagility, which makes it capable of cruising outdoor habitats in urban areas [88]. This feature has led researchers to propose that the oriental cockroach may be able to spread to new places without human intervention [88]. This may also allow this cockroach to collect diverse microorganism strains other than human pathogens, helping to understand the trend found in our sampling. Further comparative collection between coexisting synanthropic cockroaches is needed to test this idea. It is necessary to explore the microbiota of *B. orientalis* further and its relevance to the adaptation of this insect to the urban habitat.

## Figures and Tables

**Figure 1 insects-15-00903-f001:**
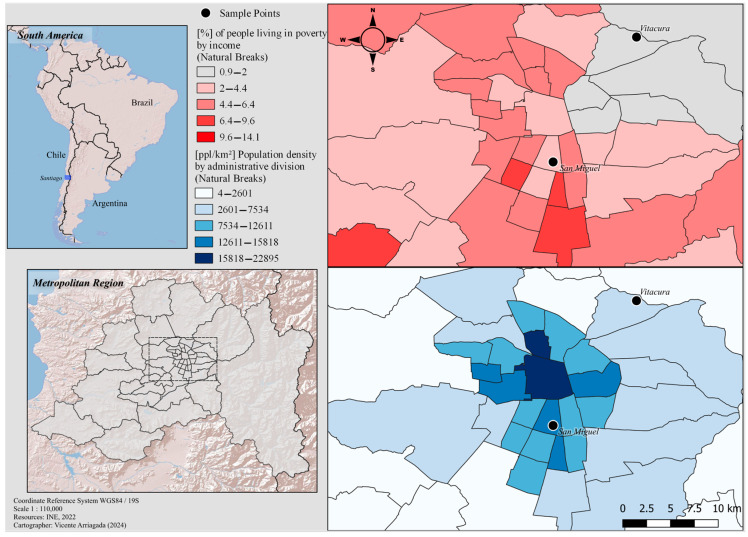
Map of the study area. The left panel shows the reference above in relation to South America and below the location corresponding to the metropolitan region. The right panel, above the map, in red scale, shows the percentage of people living in poverty, and below, in blue scale, the population density (ppl/km^2^).

**Figure 2 insects-15-00903-f002:**
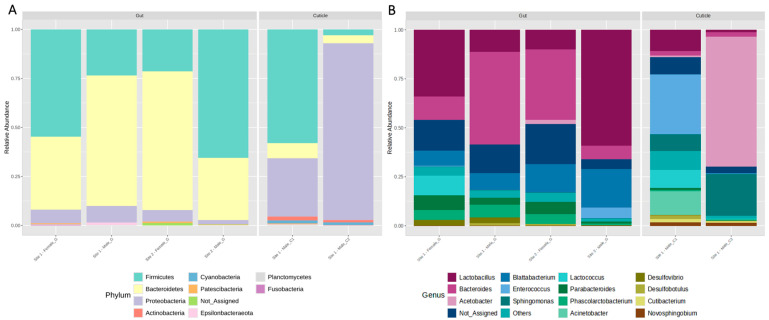
Relative abundance of taxa found in the gut and cuticle of *B. orientalis* (**A**) phylum level and (**B**) genus level. “Not assigned” corresponds to bacterial lineages that cannot be assigned to the phylum or genus level.

**Figure 3 insects-15-00903-f003:**
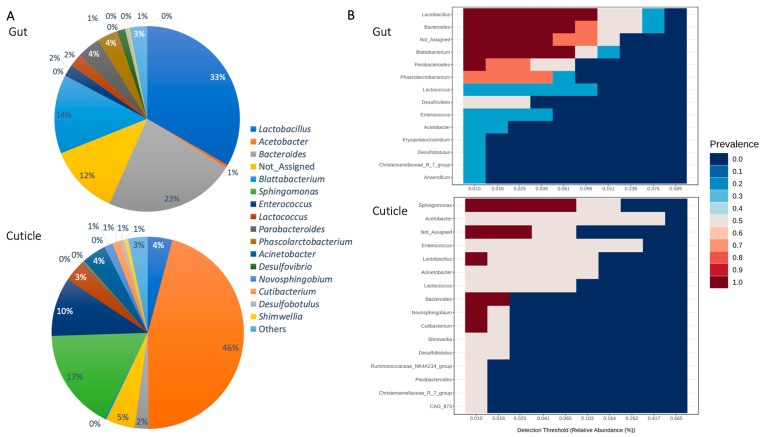
(**A**) Pie chart of the most abundant genera in the gut and cuticle samples. (**B**) Heatmap of the bacterial microbiome core as a function of abundance threshold for bacterial genera with prevalence more significant than 0.2. The X axis represents detection thresholds (indicated as relative abundance) from lowest (left) to highest (right) abundance values. “Not assigned” corresponds to bacterial lineages that cannot be assigned to the genus level.

**Figure 4 insects-15-00903-f004:**
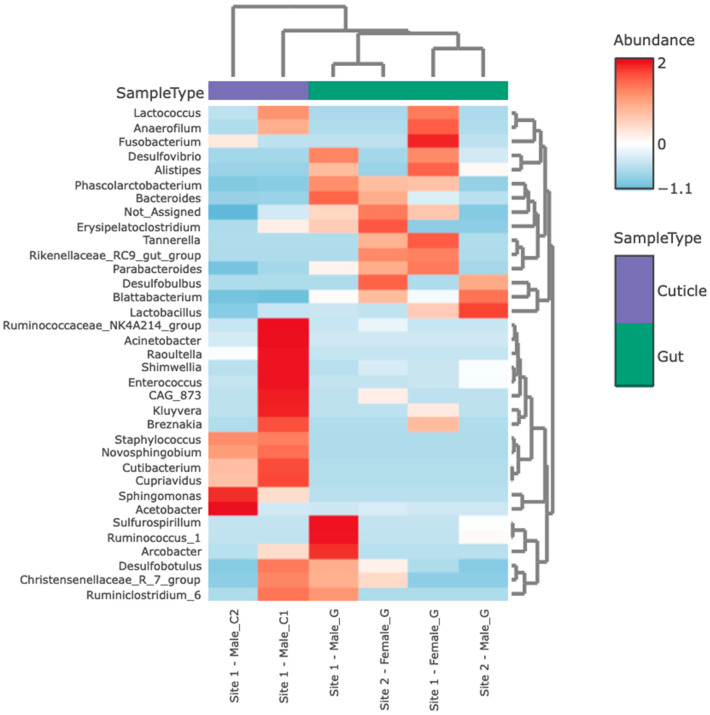
Heatmap (distance measure: Euclidean and the Hierarchical clustering algorithm), highlighting cuticle (purple) and gut (green) microbiomes clustering from *B. orientalis*. The Y axis lists strains extracted, while the X axis shows the individual specimens from which these samples were extracted.

**Figure 5 insects-15-00903-f005:**
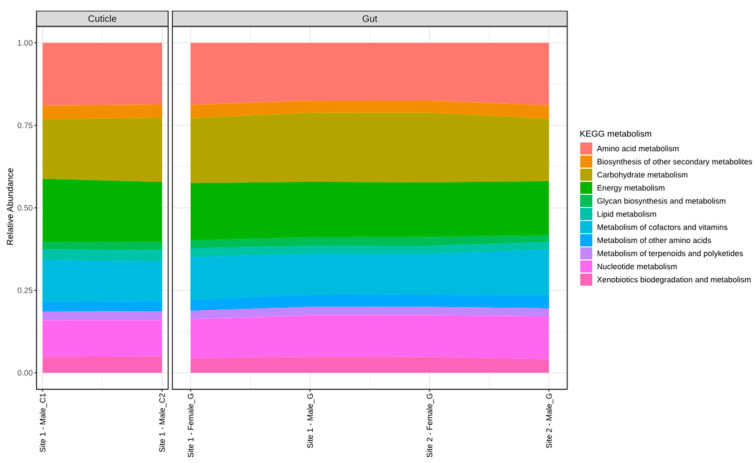
Histogram indicating the functional differences between the gut and cuticle microbiota. KEGG metabolic categories were obtained from 16S rRNA gene sequences using the Tax4Fun.

**Table 1 insects-15-00903-t001:** Bacteria associated with *B. orientalis,* according to published works. Abbreviation: HP: Human pathogen; NP: Non pathogenic.

Bacteria	Location	Association	Country	References
*Staphylococcus* sp.	Digestive tube/Cuticle	HP	Algeria	[19,22,23]
*Staphylococcus aureus*	Cuticle	HP	Algeria	[23]
*Staphylococcus epidermis*	Cuticle	HP	Iran	[24]
*Staphylococcus saprophyticus*	Cuticle	HP	Iran	[24]
*Citrobacter freundii*	Digestive tube/Cuticle	HP	Great Britain; Iran	[19,24]
*Citrobacter diversus*	Cuticle	HP	Iran	[24]
*Enterobacter cloacae*	Digestive tube	HP	Great Britain	[19]
*Enterobacter aerogenes*	Cuticle	HP	Algeria	[23]
*Escherichia coli*	Digestive tube/Cuticle	HP	Great Britain; Iran	[19,24]
*Klebsiella edward*	Digestive tube	HP	Great Britain	[19]
*Klebsiella oxytoca*	Cuticle	HP	Algeria; Iran	[23,24]
*Klebsiella ozaenae*	Digestive tube	HP	Great Britain	[19]
*Klebsiella pneumoniae*	Cuticle	HP	Iran	[24]
*Proteus vulgaris*	Digestive tube	HP	Great Britain; Algeria	[19,23,24]
*Proteus mirabilis*	Cuticle	HP	Iran	[24]
*Serratia marcescens*	Digestive tube/Cuticle	HP	Great Britain	[19]
*Serratia liquefasciens*	Cuticle	HP	Algeria	[23]
*Acinetobacter anitratus*	Digestive tube	HP	Great Britain	[19]
*Pseudomonas aeruginosa*	Digestive tube/Cuticle	HP	Great Britain; Iran	[19,24]
*Pseudomonas luteola*	Cuticle	HP	Great Britain	[23]
*Bacillus* spp.	Digestive tube/Cuticle	HP	Great Britain; Algeria; Iran	[19,23,24]
*Bacillus polymyxa*	Digestive tube	HP	Great Britain	[19]
*Bacillus circulans*	Digestive tube	HP	Great Britain	[19]
*Bacillus brevis*	Digestive tube	HP	Great Britain	[19]
*Bacillus pantothenticus*	Digestive tube	HP	Great Britain	[19]
*Bacillus pulvifaciens*	Digestive tube	HP	Great Britain	[19]
*Bacillus coagulans*	Digestive tube	HP	Great Britain	[19]
*Bacillus subtilis*	Digestive tube	HP	Great Britain	[19]
*Bacillus megaterium*	Digestive tube	HP	Great Britain	[19]
*Bacillus licheniformis*	Digestive tube	HP	Great Britain	[19]
*Bacillus firmus*	Digestive tube	HP	Great Britain	[19]
*Bacillus cereus*	Digestive tube	HP	Great Britain	[19]
*Streptococcus* spp.	Digestive tube/Cuticle	HP	Great Britain; Algeria	[19,23]
*Streptococcus bovis*	Digestive tube	HP	Great Britain	[19]
*Streptococcus equinus*	Digestive tube	HP	Great Britain	[19]
*Streptococcus durans*	Digestive tube	HP	Great Britain	[23]
*Streptococcus faecalis*	Digestive tube	HP	Great Britain	[19]
*Streptococcus faecium*	Digestive tube	HP	Great Britain	[19]
*Streptococcus sanguis*	Digestive tube	HP	Great Britain	[19]
*Streptococcus lactis*	Digestive tube	HP	Great Britain	[19]
*Streptococcus cremoris*	Digestive tube	HP	Great Britain	[19]
*Micrococcus* sp.	Digestive tube	HP	Great Britain	[19]
*Aerococcus* sp.	Digestive tube	HP	Great Britain	[23]
*Raoultella ornithinolytica*	Cuticle	HP	Algeria	[23]
*Hafnia alvei*	Cuticle	HP	Algeria	[23]
*Shimwellia blattae*	Digestive tube	NP	Algeria; Great Britain	[19,23]
*Morganella* sp.	Cuticle	HP	Iran	[24]

**Table 2 insects-15-00903-t002:** Core bacterial genera present in *B. orientalis* and their associations reported in the scientific literature.

Genera	Location	Associations	References
*Acetobacter*	Gut/Cuticle	Insect microbiota	[37,38,39]
*Phascolarctobacterium*	Gut	Human microbiota and feces	[59,60]
*Bacteroides*	Gut/Cuticle	Animal microbiota and feces	[42]
*Blattabacterium*	Gut	Cockroach microbiota	[43]
Christensenellaceae R 7 group	Gut/Cuticle	Insect microbiota	[43,44,45]
*Desulfobotulus*	Gut/Cuticle	Sediments and rhizosphere	[46,47,48]
*Desulfovibrio*	Gut	Termite microbiota	[48]
*Shimwellia*	Cuticle	Cockroach microbiota	[23]
*Enterococcus*	Gut/Cuticle	Human pathogen and insect microbiota	[49,50]
*Erysipelatoclostridium*	Gut	Human pathogen and gut microbiota	[51,52]
*Lactobacillus*	Gut/Cuticle	Animals and plants microbiota	[53]
*Acinetobacter*	Cuticle	Insect microbiota/Human pathogen	[54,55]
CAG_873	Cuticle	Animal microbiota	[56,57]
*Cutibacterium*	Cuticle	Human microbiota	[58]
*Anaerofilum*	Gut	Animal microbiota	[59,60]
Ruminococcaceae NK4A214 group	Cuticle	Human microbiota	[61]
*Novosphingobium*	Cuticle	Insect microbiota/Soil	[62,63]
*Sphingomonas*	Cuticle	Fungi antagonists/Animal pathogen	[64]
*Lactococcus*	Gut/Cuticle	Insect microbiota	[65,66]
*Parabacteroides*	Gut/Cuticle	Animal microbiota and feces	[67]

## Data Availability

The data can be found in the Appendix A.

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
