# Peer review of "Microorganism Diversity Found in Blatta orientalis L. (Blattodea: Blattidae) Cuticle and Gut Collected in Urban Environments"

_insects, 2024, doi:10.3390/insects15110903_

Round 1

Reviewer 1 Report

Comments and Suggestions for Authors

The manuscript presented the bacterial community over the exoskeleton and inside the gut of the oriental cockroach. The experimental design is very simple and clear, and the conclusions will help researchers understand the relationship between cockroach bacterial community and humans.

I have some major comments, suggesting that the author add experiments to explain or deepen the discussion.

1. The author selected two sites where there were differences in human income and population density to collect cockroaches. However, the number of samples collected at each site is small, and there are differences between male and female, which cannot reflect the true situation of the two sites. Moreover, the manuscript did not explain the collection method and microenvironment, which may affect the microbial community of the cockroach. In addition, the results did not show in detail the differences in microbial communities between the two site samples. According to the existing results, there seems to be no difference in the microbial community between the two site samples. Is this a deviation caused by too small a sample size, or human income and population density?

2. About the dendrogram analysis in figure 4, how to explain the similarities in intestinal bacterial communities among different sampling sites and different genders? Is this a deviation caused by too small a sample size?

3. The author selected some human pathogens, such as Erysipelatoclostridium ramosum, found on cockroaches for discussion. This may raise readers 'concerns about pathogenicity. It is recommended to discuss and clarify the risk of these pathogenic bacteria infecting humans through cockroaches from the aspects of bacterial transmission routes and pathogenic mechanisms to humans.

Author Response

1-The author selected two sites where there were differences in human income and population density to collect cockroaches. However, the number of samples collected at each site is small, and there are differences between male and female, which cannot reflect the true situation of the two sites. Moreover, the manuscript did not explain the collection method and microenvironment, which may affect the microbial community of the cockroach. In addition, the results did not show in detail the differences in microbial communities between the two site samples. According to the existing results, there seems to be no difference in the microbial community between the two site samples. Is this a deviation caused by too small a sample size, or human income and population density?

r1- Thank you for your observation. Regarding the sample size, this is a limitation of our work as the sequencing technique is quite expensive. Likely, this limitation is also masking more significant variability, we include this limitation in the discussion. In the published articles where they conduct surveys about the microbiota of the leading pest cockroaches, sample sizes are often reduced. For instance, less than five specimens per stage of development have been used in some cases (Chen et al., 2023; Carrasco et al., 2014). In this same sense, we would like to comment that this is the first time that a metabarcoding survey of the microbiota of Blatta orientalis has been carried out, one of the most widely distributed pest cockroach species worldwide (Nasirian, 2017), and that, without a doubt, subsequent investigations should consider extensive sampling with a larger sample size. However, to establish a line in that direction, it is necessary to be able to lay the foundations through a first brief report.

Chen, Z., Wen, S., Shen, J., Wang, J., Liu, W., & Jin, X. (2023). Composition and diversity of the gut microbiota across different life stages of American cockroach (Periplaneta americana). Bulletin of Entomological Research, 113(6), 787-793.

Carrasco, P., Pérez-Cobas, A. E., Van de Pol, C., Baixeras, J., Moya, A., & Latorre, A. (2014). Succession of the gut microbiota in the cockroach Blattella germanica. Int Microbiol, 17(2), 99-109.

Nasirian, H. (2017). Infestation of cockroaches (Insecta: Blattaria) in the human dwelling environments: a systematic review and meta-analysis. Acta Tropica, 167, 86-98.

  1. About the dendrogram analysis in figure 4, how to explain the similarities in intestinal bacterial communities among different sampling sites and different genders? Is this a deviation caused by too small a sample size?

r2- Thank you for your questionInternal communities tend to be more homogeneous, as this bacterial niche is more consistent than the variable conditions the microbiome may face. This has also been reported in other systems . We appreciate your question and discuss this pattern and the possibility that it may also be affected by a small sample size. Based on this observation we decided to remove the analysis from the paper and leave it as supplementary material, which we discuss in the discussion section acknowledging the biases in relation to the sample size.

  1. The author selected some human pathogens, such as Erysipelatoclostridium ramosum, found on cockroaches for discussion. This may raise readers 'concerns about pathogenicity. It is recommended to discuss and clarify the risk of these pathogenic bacteria infecting humans through cockroaches from the aspects of bacterial transmission routes and pathogenic mechanisms to humans.

r3: Thank you for this observation, in this new version we included further clarifications of the strains described and the extent these pathogenic bacteria may infect humans from cockroach vectors. Considering, for instance, the fact that this genus is found also in human healthy microbiota.

Reviewer 2 Report

Comments and Suggestions for Authors

Dear authors,

This interesting study highlighted the Microorganism Diversity found in Blatta orientalis (Blattodea: Blattidae) cuticle and gut collected in urban environments.The introduction needs to shortened, limiting the background of the scope of the work in the present manuscript. Also, if possible try to include culture-dependent isolation of gut bacteria which will be used for pathogenic interaction study and urban pest control.

Comments on the Quality of English Language

Can be improved

Author Response

This interesting study highlighted the Microorganism Diversity found in Blatta orientalis (Blattodea: Blattidae) cuticle and gut collected in urban environments. The introduction needs to shortened, limiting the background of the scope of the work in the present manuscript. Also, if possible try to include culture-dependent isolation of gut bacteria which will be used for pathogenic interaction study and urban pest control.

r: Thank you for your advice, in this new version we focused further the introduction. We also discuss (Discussion section, line 396) the importance of culture-dependent isolation of gut bacteria in the discussion section, as we cannot conduct these tests in this phase of our research.

Reviewer 3 Report

Comments and Suggestions for Authors

The study by Schapheer et al. details a 16S approach to define the microbiome of the cosmopolitan pest cockroach B.orientalis. The authors analyze bacterial communities from the insect exoskeleton and the gut. The authors find a range of bacterial lineages including some human pathogenic strains, which highlight a potential role of this pest species as a vector. This is an interesting finding that carry significant implications in terms of public health. The authors also attempt to examine the metabolic function of these bacterial communities via predictive analysis of KEGG terms. The experimental design and analysis is straightforward; however, a very limited and unnaturally biased selection of specimens and analytic approaches negatively impact the interpretation of the overall study. 

Major concerns:

1.    Why is the sampling uneven (lines 115-117)? Ideally, the analysis should be performed with multiple replicates of male and female specimens collected from each site. 

2.    In line 126, the authors mention that they use two females from site 1 and two males from site 2, although in lines 115-116, it is mentioned that Site 1 specimens consist of three adult males and one adult female and Site 2 specimens consist of one adult male and one adult female. This is also different from the figures which say that the gut analysis was performed with one male and one female pair from each site. 

3.    In line 132, the authors mention that for cockroach cuticle, they use two male specimens from site 1 for “gut” analysis. There is an obvious conflict of information here. But also, why the bias? The sampling seem random. The analysis should be conducted with multiple male and female specimens from each site. As it is apparent from the data presented in Figure 2B, that more sampling is needed since the patterns appear quite variable.

4.    In lines 177-189, the authors mention that Acetobacter was the most abundant genera, although from the data shown it is clear that this is biased due to one sample, which is clearly as an outlier (same site, same part of the insect). The authors also curiously do not discuss the fact that the cuticle samples are significantly different from each other in the discussion of their results although it may reflect true dynamic signature of the microbial community. The only way to know is to sample more. 

5.    In Figure 3A, why are the colors different for the same OTUs across the gut and cuticle composition pie charts? This is incredibly confusing since it makes it very difficult to compare composition across these two niches. 

6.    In Figure 3B, the authors are recommended to maintain the order of the genera between the top and bottom plots, keeping the tissue-specific/unique genera at the bottom. 

7.    Figure 4 does not show the similarity of OTUs between these samples/specimens. Please consider showing a heatmap of distance matrix values as a proportion of shared OTUs via pairwise associations.

8.    In lines 208-211, the authors perform KEGG analysis by predicting functional pathways using the KEGG ortholog database. The data presented in Figure 5 show very generalized GO terms, with no discernable differences between samples and is not very informative. What is the degree of redundancy between these communities? This will dictate how stable or dynamic the community is likely to be. This information warrants its own figure and discussion. 

9.    Re Figure 5 and KO analysis. The authors should also probably exclude Acetobacter from the KO analysis given its skewed abundance in one of the two cuticle-samples.

10. In Line 209, the authors mention that they found statistically significant differences. Between which groups/samples? Were the authors comparing between cuticle vs gut microbe-associated KO terms? If so, how are the authors controlling for variables like sex and microhabitat? One concern is that given uneven sampling, the authors’ limited dataset is underpowered to run this analysis and derive any meaningful conclusions, even at a “coarse grain scale”. 

11. In line 263, the authors conclude that in relation to microorganisms non-pathogenic to humans, that bacteria constitute the majority of the community and core microbiome of the gut. Did the authors find any archaea in the community? Since the authors only perform 16S sequencing (lines 141-142) and not metagenomic or ITC (cannot detect fungi and viruses), they cannot make this blanket statement. 

12. Table 1 is informative and a good addition since it provides context. Is there no information available for non-pathogenic bacteria associated with B. orientalis from these studies? Since all previous data originate from Europe, Asia and Africa, it would be interesting to see how the human pathogens associated with B. orientalis varies across geographical locations, especially since this study adds valuable data from a previously unsampled part of the world. Especially since the taxa recovered in this study appear to be unique to this region.

Comments on the Quality of English Language

A thorough proof-reading of the entire manuscript is recommended. There are several typographical errors.

Author Response

The study by Schapheer et al. details a 16S approach to define the microbiome of the cosmopolitan pest cockroach B.orientalis. The authors analyze bacterial communities from the insect exoskeleton and the gut. The authors find a range of bacterial lineages including some human pathogenic strains, which highlight a potential role of this pest species as a vector. This is an interesting finding that carry significant implications in terms of public health. The authors also attempt to examine the metabolic function of these bacterial communities via predictive analysis of KEGG terms. The experimental design and analysis are straightforward; however, a very limited and unnaturally biased selection of specimens and analytic approaches negatively impact the interpretation of the overall study

R: Thank you for your reviewing and suggestions. In response to reviewers’ comments, we removed the cluster analysis and dendrogram from the paper (Figure 4). In this version, we added a heatmap instead. Moreover, we briefly discuss the results, acknowledging sampling biases and a low sample size. We, therefore, focus on the aspects highlighted by the reviewer (discovery of pathogens and potential of B. orientalis as a vector), focusing on the description of the bacterial communities found.

Major concerns:

  1. Why is the sampling uneven (lines 115-117)? Ideally, the analysis should be performed with multiple replicates of male and female specimens collected from each site. 

R: Thank you for your observation. We understand this is not an ideal situation. This work is the first approach to the description of the microbiota of B. orientalis, and in this regard, we focus exclusively on the characterization of the bacterial community from a descriptive approach. Therefore, we remove the cluster analysis, since due to sampling, its interpretation is biased.

  1. In line 126, the authors mention that they use two females from site 1 and two males from site 2, although in lines 115-116, it is mentioned that Site 1 specimens consist of three adult males and one adult female and Site 2 specimens consist of one adult male and one adult female. This is also different from the figures which say that the gut analysis was performed with one male and one female pair from each site. 

R: We fixed this mistake in this version. Thank you for your observation

  1. In line 132, the authors mention that for cockroach cuticle, they use two male specimens from site 1 for “gut” analysis. There is an obvious conflict of information here. But also, why the bias? The sampling seems random. The analysis should be conducted with multiple male and female specimens from each site. As it is apparent from the data presented in Figure 2B, that more sampling is needed since the patterns appear quite variable.

r: Thank you for commenting. In this version, we make these differences more explicit and discuss the limitations of our sampling.

  1. In lines 177-189, the authors mention that Acetobacter was the most abundant genera, although from the data shown it is clear that this is biased due to one sample, which is clearly as an outlier (same site, same part of the insect). The authors also curiously do not discuss the fact that the cuticle samples are significantly different from each other in the discussion of their results although it may reflect true dynamic signature of the microbial community. The only way to know is to sample more. 

R: In this version, we address the issue of cuticle differences in bacteria diversity, making explicit these differences and the constraint of reduced sample size.

  1. In Figure 3A, why are the colors different for the same OTUs across the gut and cuticle composition pie charts? This is incredibly confusing since it makes it very difficult to compare composition across these two niches. 

R: Thank you for your suggestion, we corrected the use of color in coherence with your recommendation.

  1. In Figure 3B, the authors are recommended to maintain the order of the genera between the top and bottom plots, keeping the tissue-specific/unique genera at the bottom. 

R: Regarding the core microbiome analysis in Microbiome Analyst, the order of appearance of the genera is defined by the importance of the genus in the habitat analyzed; therefore, despite recognizing this issue, we cannot modify this presentation of results.

  1. Figure 4 does not show the similarity of OTUs between these samples/specimens. Please consider showing a heatmap of distance matrix values as a proportion of shared OTUs via pairwise associations.

R: We appreciate this suggestion, we implemented the heatmap in this version.

  1. In lines 208-211, the authors perform KEGG analysis by predicting functional pathways using the KEGG ortholog database. The data presented in Figure 5 show very generalized GO terms, with no discernable differences between samples and is not very informative. What is the degree of redundancy between these communities? This will dictate how stable or dynamic the community is likely to be. This information warrants its own figure and discussion. 

R:. We appreciate your suggestion. Nonetheless, our sample size restricts our ability to conduct redundancy comparisons. In any case, we provide the whole dataset as supplementary material.

  1. Re Figure 5 and KO analysis. The authors should also probably exclude Acetobacter from the KO analysis given its skewed abundance in one of the two cuticle-samples.

R: Thank you for your comment. In this version, we report the skewed abundance of bacteria strains from cuticle samples more explicitly. However, since this is the first description of the bacterial diversity of B. orientalis microbiota assessed by metabarcoding, we preferred not to exclude any lineages. We included all the details in the supplementary material.

  1. In Line 209, the authors mention that they found statistically significant differences. Between which groups/samples? Were the authors comparing between cuticle vs gut microbe-associated KO terms? If so, how are the authors controlling for variables like sex and microhabitat? One concern is that given uneven sampling, the authors’ limited dataset is underpowered to run this analysis and derive any meaningful conclusions, even at a “coarse grain scale”.

R: Thank you for this observation. Due to the reduced sample size, this could be a problem. In this version, we eliminated this analysis and did not consider it in the results (Table S3 deleted).

  1. In line 263, the authors conclude that in relation to microorganisms non-pathogenic to humans, that bacteria constitute the majority of the community and core microbiome of the gut. Did the authors find any archaea in the community? Since the authors only perform 16S sequencing (lines 141-142) and not metagenomic or ITC (cannot detect fungi and viruses), they cannot make this blanket statement. 
  2. We appreciate your suggestion. In this version, we restrained our commentaries to the resolution of the analyses applied. We make explicit in the introduction and discussion that in this analysis we only included bacteria.
  3. Table 1 is informative and a good addition since it provides context. Is there no information available for non-pathogenic bacteria associated with B. orientalis from these studies? Since all previous data originate from Europe, Asia and Africa, it would be interesting to see how the human pathogens associated with B. orientalis varies across geographical locations, especially since this study adds valuable data from a previously unsampled part of the world. Especially since the taxa recovered in this study appear to be unique to this region.

R: Most available information published on B. orientalis bacteria is regarding human pathogens. The focus of previous work was on pathogenic strains (with the exception of Shimwellia blattae). We comment this in the discussion section. This fact drives us to conduct our sampling considering the whole associated communities, as despite non-pathogenic may not directly impact humans, these may contribute to cockroach survival in urban environments. Considering your observation of the geographical variability of pathogenic strains, we discuss further these patterns in this new version of our ms.